# *Trichoderma* versus *Fusarium*—Inhibition of Pathogen Growth and Mycotoxin Biosynthesis

**DOI:** 10.3390/molecules27238146

**Published:** 2022-11-23

**Authors:** Marta Modrzewska, Lidia Błaszczyk, Łukasz Stępień, Monika Urbaniak, Agnieszka Waśkiewicz, Tomoya Yoshinari, Marcin Bryła

**Affiliations:** 1Department of Food Safety and Chemical Analysis, Waclaw Dabrowski Institute of Agricultural and Food Biotechnology—State Research Institute, Rakowiecka 36, 02-532 Warsaw, Poland; 2Plant Microbiomics Team, Institute of Plant Genetics, Polish Academy of Sciences, 60-479 Poznan, Poland; 3Plant-Pathogen Interaction Team, Institute of Plant Genetics, Polish Academy of Sciences, 60-479 Poznan, Poland; 4Department of Chemistry, Poznan University of Life Sciences, Wojska Polskiego 75, 60-625 Poznan, Poland; 5Division of Microbiology, National Institute of Health Sciences, 3-25-26 Tonomachi, Kawasaki-ku, Kawasaki-shi 210-9501, Kanagawa, Japan

**Keywords:** *Trichoderma* spp., *Fusarium* spp., mycotoxins, plant biocontrol

## Abstract

This study evaluated the ability of selected strains of *Trichoderma viride*, *T*. *viridescens*, and *T*. *atroviride* to inhibit mycelium growth and the biosynthesis of mycotoxins deoxynivalenol (DON), nivalenol (NIV), zearalenone (ZEN), α-(α-ZOL) and β-zearalenol (β-ZOL) by selected strains of *Fusarium culmorum* and *F*. *cerealis*. For this purpose, an in vitro experiment was carried out on solid substrates (PDA and rice). After 5 days of co-culture, it was found that all *Trichoderma* strains used in the experiment significantly inhibited the growth of *Fusarium* mycelium. Qualitative assessment of pathogen–antagonist interactions showed that *Trichoderma* colonized 75% to 100% of the medium surface (depending on the species and strain of the antagonist and the pathogen) and was also able to grow over the mycelium of the pathogen and sporulate. The rate of inhibition of *Fusarium* mycelium growth by *Trichoderma* ranged from approximately 24% to 66%. When *Fusarium* and *Trichoderma* were co-cultured on rice, *Trichoderma* strains were found to inhibit DON biosynthesis by about 73% to 98%, NIV by about 87% to 100%, and ZEN by about 12% to 100%, depending on the pathogen and antagonist strain. A glycosylated form of DON was detected in the co-culture of *F*. *culmorum* and *Trichoderma,* whereas it was absent in cultures of the pathogen alone, thus suggesting that *Trichoderma* is able to glycosylate DON. The results also suggest that a strain of *T*. *viride* is able to convert ZEN into its hydroxylated derivative, β-ZOL.

## 1. Introduction

It is estimated that plant diseases contribute to 15% of crop losses worldwide, and about 80% of these losses are caused by pathogenic fungi [1,2]. In recent years, fungal diseases of cereal plants have become a serious threat as a consequence of climate change [3]. *Fusarium* fungi are among the most important pathogens contributing to crop yield loss and reduced grain quality [4]. The greatest losses in cultivated cereal crops are associated with fusarium head blight (FHB) or fusarium ear rot (FER) in maize, which are mainly caused by *F*. *graminearum* and *F*. *culmorum* [5]. These fungi are common in Europe, North America, and Asia. It is thought that the temperatures and water activity (aw) optimal for *Fusarium* growth are 20–30 °C and 0.980–0.995, respectively. The optimal conditions for *F*. *graminearum* growth are 25–30 °C and aw = 0.98, while for *F*. *culmorum* 20–25 °C and aw > 0.98 [6]. Infection of cereal plants by *Fusarium* is often associated with the biosynthesis of mycotoxins. The most common mycotoxins include trichothecenes (e.g., deoxynivalenol (DON), nivalenol (NIV), and zearalenone (ZEN)) [7,8,9]. DON and NIV exhibit immunotoxic, cytotoxic, and genotoxic effects, and NIV additionally exhibits hepatotoxic effects [10,11,12]. ZEN, on the other hand, can disrupt the function of the endocrine system and contribute to genital lesions due to oestrogen receptor binding, posing a risk to human and animal health [13].

In recent years, the food laws related to the use of pesticides in plant protection has become extremely restrictive in the European Union. Farmers are obliged to produce their crops according to integrated procedures, including biological methods in plant protection [14,15]. Additionally, in the context of current climate change, the migration of pathogens and pests towards the poles is considered a challenge [16]. This can result in changes in pathogen populations responsible for cereal plant infections. Residues of plant protection products in agricultural raw materials may be dangerous for human health also when present in low, permitted amounts. In view of these challenges, alternative, environmentally friendly measures of plant protection against cereal plant pathogens are needed [17]. Recently, the potential of microorganisms exhibiting antagonistic properties against various pathogens has received increasing attention. Plant protection by means of microorganisms can offer many advantages as these not only compete with pathogens, but also contribute to improved plant performance through their biostimulatory action [18]. Of particular interest among the vast group of potential biocontrol microorganisms, are *Trichoderma* fungi. These are found in all climate zones, residing in a variety of environments, most commonly in the soil [19]. As reported by Zehra et al. [20], the optimal conditions for the growth and spore production by *Trichoderma* fungi are temperatures ranging from 25–40 °C and a water activity as high as 0.98 [6,21]. Many *Trichoderma* species have been used as agents to improve the efficiency of nutrient use by crop plants [22]. *Trichoderma* spp. display diverse biostimulatory activities and antagonistic properties. Antagonistic properties of *Trichoderma* fungi include the ability to compete for space and nutrients, mycoparasitism, and antibiosis [23].

Recent studies have proven that *Trichoderma* fungi are not only capable of displaying antagonistic properties towards other pathogenic fungi but can also inhibit the biosynthesis of mycotoxins [24]. Furthermore, enzymes secreted by these fungi may promote mycotoxin degradation or biotransformation [25]. Mycotoxin modifications involving fungal enzymes can lead to the formation of new compounds with varying in toxicity compared to their basic analogues [26]. Biocontrol measures are currently underexplored in the context of limiting *Fusarium* growth in cereal crops [27]. Therefore, it is necessary to pre-select different strains of individual *Trichoderma* spp. to assess their antagonistic potential. This paper presents an assessment of the antagonistic capacity of selected *Trichoderma* strains against *F*. *culmorum* and *F*. *cerealis* strains, and of their ability to inhibit the biosynthesis of *Fusarium*-produced toxins.

## 2. Results

### 2.1. Evaluation of the Antagonistic Potential of Trichoderma Fungi

Antagonistic tests showed that all *Trichoderma* strains inhibited the growth of the tested *F*. *culmorum* and *F*. *cerealis* strains (Figure 1 and Figure 2). After 5 days of incubation, the pathogen inhibition rate ranged between 24% and 66%, depending on the pathogen and the *Trichoderma* strain (Table 1).

Qualitative assessment of the interactions between colonies after 5 days of co-cultures of *Trichoderma* spp. and *Fusarium* spp. are presented in Table 2. All the studied *Trichoderma* strains outgrew and sporulated on the mycelium of *F*. *culmorum* KF191. *T*. *atroviride* AN523 also overgrew the mycelium of *F*. *culmorum* KF846. This strain was the most effective antagonist against *Fusarium* spp. In the qualitative assessment it received the highest average score of 7 in the Mańka’s scale.

### 2.2. The Effect of Trichoderma on Mycotoxin Biosynthesis by Fusarium *spp.*

The ability of *Trichoderma* to inhibit the biosynthesis of *Fusarium* mycotoxins was assessed. The presence of DON, 3- + 15-acetyl-deoxynivalenol (3- + 15-AcDON), deoxynivalenol-3-glucoside (DON-3G), NIV, NIV-3G, fusarenone X (FUS-X), ZEN, zearalenone-14-sulphate (ZEN-14S), zearalenone-14-glucoside (ZEN-14G), α-zearalenol (α-ZOL) and β-zearalenol (β-ZOL) was analysed. These mycotoxins were present in the post-culture media (Table 3). The presence of ZEN-14G or NIV-3G (>LOQ) was not confirmed in any of the tested samples.

*F*. *culmorum* KF191 strain produced 3- + 15-AcDON, ZEN, ZEN-14S, α-ZOL, and β-ZOL in the control culture at concentrations of 23.95, 47.36, 3.608, 0.50, and 1.28 mg/kg, respectively. In all bi-culture combinations, *Trichoderma* strains significantly reduced the levels of the tested toxins, which ranged from below LOQ to 2.85 mg/kg (for 3- + 15-AcDON), from 0.56 to 28.21 mg/kg (for ZEN), from 1.78 to 90.78 mg/kg (for ZEN-14S), from below LOQ to 0.18 mg/kg (for α-ZOL) and from 0.07 to 1.05 mg/kg (for β-ZOL).

*F*. *culmorum* KF350 strain was able to synthesize NIV, FUSX, ZEN, ZEN-14S, α-ZOL, and β-ZOL at the levels of 15.83, 86.86, 12.20, 641.68, and 0.28 0.62 mg/kg, respectively. Only the bi-culture combination KF350 vs. AN355 had a significantly higher β-ZOL content (i.e., 1.09 mg/kg), while the other combinations exhibited a significantly lower content of this compound (i.e., 0.21–0.74 mg/kg). For other toxins their levels in the bi-cultures were significantly lower than in the control medium, ranging from below LOQ to 0.31 mg/kg (for NIV), 0.13 to 5.18 mg/kg (for FUS-X), 0.25 to 10.75 mg/kg for (ZEN), 17.31 to 57.61 mg/kg (for ZEN-14S) and from below LOQ to 0.22 mg/kg (for α-ZOL).

DON, 3- + 15-AcDON, ZEN, ZEN-14S α-ZOL, and β-ZOL were identified in the control *F*. *culmorum* KF846 strain at average levels of 82.53, 111.09, 164.29, 2714.08, 2.17 and 2.77 mg/kg, respectively. Additionally, in the KF846 vs. AN690 combination a higher β-ZOL content (4.60 mg/kg) was observed than in the control KF846 strain. The presence of DON-3G was observed in the co-culture medium, whereas it was not found in the control medium. DON-3G content in these media ranged from below LOQ to 2.43 mg/kg, depending on the combination For other toxins, the contents in the bi-culture media were significantly lower than control sample (*F*. *culmorum* KF846) (Table 3).

The KF583 *F*. *cerealis* strain synthesized NIV, FUS-X, ZEN, ZEN-14S, and β-ZOL in control culture, at the levels of 24.07, 119.70, 238.63, 9337, and 6.63 mg/kg, respectively. The levels of NIV, FUS-X, ZEN, ZEN-14S, and β-ZOL were also lower in media from co-culture than in the control sample (*F*. *cerealis* KF583), and ranged from below LOQ to 0.84, from below LOQ to 11.98, from 0.31 to 15.86, from 0.31 to 2.20, from 0.67 to 9.70 and from below LOQ to 0.83 mg/kg, respectively.

## 3. Discussion

### 3.1. Antagonistic Properties

*Trichoderma* spp. exhibit antagonistic properties towards other fungi and microorganisms, mostly through competition, antibiosis, and mycoparasitism [28]. *Trichoderma* fungi are among the most aggressive competitors of *Fusarium*, and mycoparasitism is their key biocontrol mechanism [29,30]. In our experiment it was shown that all strains of *T*. *viride*, *T*. *viridescens,* and *T*. *atroviride* tested were capable of inhibiting the growth of *F*. *culmorum* and *F*. *cerealis* with an average inhibition rate of 33% to 54%, depending on the *Trichoderma* strain. So far, published research has shown the potential of *Trichoderma* fungi to limit the growth of pathogenic *Fusarium* species and the results were consistent with our findings. Błaszczyk et al. [31] observed that *T*. *atroviride*, *T*. *citrinoviride*, *T*. *cremeum*, *T*. *hamatum*, *T*. *harzianum*, *T*. *koningiopsis*, *T*. *longibrachiatum*, *T*. *longipile*, *T*. *viride,* and *T*. *viridescens* inhibited the mycelium growth of *F*. *culmorum* in vitro and the inhibition rate ranged from 27% to 80%. On the other hand, Larran et al. [32] showed that *T*. *harzianum* in a co-culture with *F*. *graminearum* was able to inhibit mycelium growth of the pathogen in vitro by 46%. Relatively high (up to 81%) inhibition of *F*. *graminearum* growth by *T*. *citrinoviride*, *T*. *harzianum,* and *T*. *asperellum* was obtained in vitro by Xue et al. [33]. The discrepancies between ours and published results were probably caused by differences in experimental models used. Similarly, Veenstra et al. [34] observed a 52% growth inhibition of *F*. *verticillioides* by *T*. *asperellum* after simultaneous strain inoculation on a PDA medium, probably caused by competition for resources, as *T*. *asperellum* grew faster than *F*. *verticillioides*. As reported by Perincherry et al. [35], the faster growth of *Trichoderma* species may be due to competition for nutrients. Metabolites putatively produced by *Trichoderma* can also be a limiting factor for the growth of pathogenic *Fusarium* species. Mironenka et al. [36] found that some metabolites biosynthesized by *T*. *harzianum* (i.e., 14-amino acid peptaibols, T22-azophilone and harzianic acid) caused severe oxidative stress and inhibited the growth of *F*. *culmorum*.

In our study all *Trichoderma* strains studied outgrew and sporulated on the *F*. *culmorum* KF191 mycelium and *T*. *atroviride* AN523 also outgrew the *F*. *culmorum* KF846 mycelium. Sallam et al. [37] found that *T*. *atroviride* and *T*. *longibrachiatum* wrap themselves around the hyphae of the pathogen to inhibit the growth of *F*. *oxysporum* and produced substances which may play an important role in the lysis of cell wall components of the pathogen. The ability of *Trichoderma* to wrap around or penetrate the mycelium of a pathogen was confirmed by Halifu et al. [38] in a co-culture of *T*. *virens* and *R*. *solani*. Matarese et al. [39] also highlighted mycoparasitism, mediated by the production of cell wall-degrading enzymes (CWDEs) as a key biocontrol mechanism for *Trichoderma* strains.

### 3.2. Biosynthesis Inhibition of Fusarium Toxins by Trichoderma Species

The studied *Trichoderma* strains inhibited mycotoxin biosynthesis by *Fusarium* fungi (DON, 3- + 15-AcDON, NIV, FUS-X, ZEN, α-ZOL, β-ZOL, and ZEN-14S) by 12% to 100% relative to the control samples. This inhibition was probably caused by a reduction in the size of pathogenic mycelium and/or the interference of *Fusarium* and *Trichoderma* metabolic pathways. Similar studies were carried out [31,36,40] and very similar results were obtained by Tian et al. [40], who observed in bi-cultures of *F*. *graminearum* and various *Trichoderma* strains that ZEN biosynthesis was inhibited by 9% to 97%, α-ZOL by 31% to 87% and β-ZOL by 34% to 89%. In our experiment, the biosynthesis of mycotoxins tested was inhibited by 12% to 99%, 23% to 100%, and 13% to 100%, respectively. Mironenka et al. [36] also observed that the presence of *T*. *harzianum* in co-culture with *F*. *culmorum* resulted in the inhibition of ZEN biosynthesis by more than 90%. Our findings were consistent with previous reports of Błaszczyk et al. [31], who showed that the inhibition of *F*. *culmorum* and *F*. *cerealis* mycotoxin biosynthesis by *T*. *atroviride* and *T*. *viride* was correlated with the inhibition of pathogen mycelium growth. Studied *Trichoderma* strains inhibited DON biosynthesis at an average level of 75% to 92%, 72% to 96% for NIV, 21% to 99% for ZEN, and 77% to 100% for 3-AcDON.

Lower mycotoxin levels in the co-cultured media compared to *Fusarium* monocultures seem to be a natural consequence of mycelium inhibition of the pathogen. However, an almost two-fold increase of β-ZOL (1.1 and 4.6 mg/kg) was observed in *F*. *culmorum* KF350 and *F*. *culmorum* KF846 samples co-cultured with *T*. *viride* AN355. The above observation may be related to the initiation of unspecified stress factors in *Fusarium* by the antagonist. The available literature reports little information indicating that the presence of a competitor (*Trichoderma* spp.) stimulates pathogen mycotoxin biosynthesis. However, as reported by Voigt et al. [41], external stresses, such as competing microorganisms can activate defence mechanisms in pathogenic *Fusarium* species and thus contribute to increased pigments (aurofusarin and rubrofusarin), microsporins and mycotoxin (DON, ZEN) biosynthesis. Mironenka et al. [42] suggested that metabolites biosynthesized by *Trichoderma* caused severe oxidative stress and affected ZEN production. The reason for the higher β-ZOL level in the co-culture medium compared to the monocultures can also be explained by the ability of the *Trichoderma* fungi to biotransform ZEN to β-ZOL. ZEN is a toxin that is always present together with β-ZOL at significant concentrations. As confirmed by Tian et al. [40], *T*. *atroviride* in a ZEN-containing medium (2 µg/mL) was marginally able to biotransform the toxin to the α-ZOL and β-ZOL forms [40]. In our results, ZEN-14S was observed in monocultures with *Fusarium* at very high levels (641.7–9.337 mg/kg) compared to ZEN levels (12.2–238.6 mg/kg). We believe that the studied *Fusarium* strains may be able to activate defence mechanisms against the toxic effects of accumulated ZEN. One of these mechanisms may involve the ZEN sulphation reaction as our previous studies suggest [43]. Tian et al. [40] believed that ZEN sulphation is one of the detoxification processes of this compound by *Trichoderma*. These conclusions were observed after cultivating cultures of *T*. *harzianum*, *T*. *koningii*, *T*. *longibrachiatum*, *T*. *atroviride*, *T*. *asperellum,* and *T*. *virens* on a ZEN-containing medium. Although the authors demonstrated the presence of ZEN-14S in the substrate, no quantitative analysis of this substance was performed. In our study lower levels of both ZEN-14S and ZEN were observed in co-cultures, with ZEN-14S prevailing. The hypothesis that *Trichoderma* fungi is able to biotransform ZEN to ZEN-14S remains speculative, as the presence of ZEN-14S in the substrates may be the result of the combined activities of pathogen and antagonist.

A very interesting observation was made when *F*. *culmorum* KF846 was co-cultured with *T*. *viride* AN355, *T*. *viride* AN690, *T*. *viridescens* AN508, *T*. *viridescens* AN609, *T*. *atroviride* AN705 and *T*. *atroviride* AN153 with DON-3G being detected. This compound was not observed in the pathogen monoculture, which could indicate the ability of *Trichoderma* to biotransform DON via glycosylation. Thus far, very little is known about the capacity of *Trichoderma* fungi to glycosylate mycotoxins [40,44]. Tian et al. [44] showed that *T*. *harzianum*, *T*. *atroviride,* and *T*. *asperellum* strains contributed to the inhibition of DON biosynthesis and its biotransformation to DON-3G in contact with *F*. *graminearum*. Although the ability of the *Trichoderma* fungi to transform DON to DON-3G was confirmed in our study, this ability was not demonstrated in the case of NIV. Presumably, this could be caused by the lower levels of NIV (from below the LOQ level to 2.01 mg/kg) in the co-culture media compared to the levels of DON (from below the LOQ level to 22.34 mg/kg). While the ability of *Trichoderma* fungi to glycosylate DON was proven, it was not observed for ZEN. ZEN-14G (above the LOQ level) was not observed in control media nor in media with co-cultured strains. Our results were consistent with the observations of Tian et al. [40], who also did not find ZEN-14G in the media. This particular aspect of the study will be researched further. The findings and their implications should be discussed in the broadest context possible. Future research directions may also be highlighted.

## 4. Materials and Methods

### 4.1. Chemicals and Reagents

Solutions of analytical standards DON, DON-3G, 3-acetyl-deoxynivalenol (3-AcDON), 15-acetyl-deoxynivalenol (15-AcDON), NIV, FUSX, ZEN, α-ZOL, β -ZOL were purchased from Romer Labs (Tulln, Austria) and ZEN-14S and ZEN-14G were purchased from Aokin (Berlin, Germany). The certified standard of NIV-3G (110 µg/mL) was isolated from wheat according to the procedure described by Yoshinari et al. [45]. Methanol and water of LC-MS purity were purchased from Witko (Lodz, Poland), formic acid 98–100% (ultrapure) and ammonium formate 97% (extra pure) were purchased from Chem-Lab (Zedelgem, Belgium). Potato-dextrose agar (PDA) was provided by Oxoid (Basingstoke, UK).

### 4.2. Fungal Isolates

Four strains of *Fusarium* spp. (i.e., *F*. *culmorum* KF191, *F*. *culmorum* KF350, *F*. *culmorum* KF846, *F*. *cerealis* KF583) and eight strains of *Trichoderma* spp. (i.e., *T*. *viridescens* AN508, *T*. *viridescens* AN609, *T*. *viride* AN355, *T*. *viride* AN690, *T*. *atroviride* AN153, *T*. *atroviride* AN215, *T*. *atroviride* AN523 and *T*. *atroviride* AN705) were used in the studies. All the species were obtained from the culture collection of the Institute of Plant Genetics of the Polish Academy of Sciences in Poznań, Poland.

### 4.3. Dual Culture Bioassay

Experiments were designed to evaluate the inhibition of pathogenic fungi growth and conducted in bi-cultures, on an agar medium (potato dextrose agar, PDA) according to the method described by Błaszczyk et al. [31]. The fungal culture was grown on 8.5 cm diameter Petri dishes. The fungi were inoculated simultaneously, in a combination: antagonistic strain with pathogen strain on the opposite sides of the plate at a distance of approximately 8 cm between the strains. Single cultures on PDA plates were the control samples. Plates were incubated at 25 ± 2 °C, 12 h/12 h night/day cycle. The cultures were grown for 5 days. The fungal growth was measured with a ruler, every 24 h, up to the day of contact between the mycelium of the pathogen and the antagonist. The experiment was conducted in independent triplicates.

After 5 days of co-incubation, a qualitative assessment of pathogen–antagonist interactions was carried out using the scale proposed by Mańka [46]. The scale shows the degree of domination of one colony by another and takes the following values: 0 (the antagonist mycelium has outgrown 50% of the plate surface area), +4 (the antagonist mycelium has outgrown 75% of the plate surface area), +6 (the antagonist mycelium has outgrown 85% of the plate surface area), +8 (the antagonist mycelium has outgrown 95% of the plate surface area and/or the pathogen mycelium).

The inhibitory effect of *Trichoderma* isolates on pathogen growth was assessed by estimating the percentage (%) reduction in pathogen growth in the presence of the antagonist, according to the following formula: (Rc − R)/Rc × 100, where Rc is the estimates of radial growth of a pathogen in the control sample and R is the estimates of radial growth of a pathogen in the bi-culture [31].

### 4.4. Determination of the Mycotoxin Biosynthesis Inhibition

The fungi were inoculated simultaneously in combination: *Trichoderma* and pathogen strains on rice grains. Fungal strains growing in single cultures comprised the control samples. 15 g of long-grain white rice and 4 mL of sterile water were added to an Erlenmeyer flask, left overnight and sterilized in an autoclave the next day, at 121 °C for 30 min. The rice was inoculated with 3 cm^2^ of 7-day-old mycelium grown on a PDA medium. The average humidity of the culture was kept at approximately 30% and maintained for 14 days. After incubation, the cultures were dried at room temperature and ground using a knife-type mill (Grindomix GM 200, Retsch, Haan, Germany). The ground samples of inoculated rice were stored under freezing conditions (below −30 °C) until mycotoxin analysis.

### 4.5. Sample Preparation

The samples were prepared according to the modified method described by Uwineza et al. [47]. Ground 0.5 g samples of inoculated rice were extracted with 20 mL of an acetonitrile mixture: 0.01% formic acid (84:16, *v*/*v*) using a homogenizer (Unidrive X 1000, Cat Scientific, Paso Robles, CA, USA) at 5000 rpm for 2 min. The mixture was then centrifuged for 10 min (MPW-380R, MPW Med. Instruments, Warsaw, Poland) at 10,000 rpm. The extract was then filtered through a 0.45 µm nylon syringe filter. 2 mL of the filtrate was transferred to a round-bottom flask and evaporated in a vacuum evaporator (Heidolph Instruments, Schwabach, Germany). The residue was dissolved in 0.6 mL of methanol and 0.4 mL of 0.25% formic acid. After each solvent addition, the samples were subjected to sonication and the total volume was filtered through a nylon syringe filter (0.22 µm). If necessary (samples exceeding the analytical method range), samples were diluted with a solution mixture of the same solvent composition.

### 4.6. UHPLC-HESI-MS/MS

Mycotoxins were analysed using a liquid chromatography-Q-Exactive Orbitrap mass spectrometry setup operating with a heated electrospray interface (UHPLC-HESI-MS/MS) (Thermo Fisher Scientific, Waltham, MA, USA). The analytes were separated on a C18 Cortecs chromatography column (100 mm × 2.1 mm × 1.6 μm, Waters) with a built-in filter placed upstream of the chromatography column. The mobile phase consisted of water and methanol in ratios of 90:10 (phase A) and 10:90 (phase B), respectively. Both phases contained 5 mM ammonium formate and 0.2% formic acid. The following flow gradient was used: from 0 to 2 min, 100% phase A; from 2 to 3 min, 75% phase A; from 3 to 6 min, 40% phase A; from 6 to 26 min, 0% phase A; from 26–30 min, 100% phase A. The flow rate was 300 µL/min, and the volume of the injected sample was −2 µL. The 70,000-resolution mass spectrometer operated in both positive and negative ionisation modes, with a scan range of 100 to 1000 m/z. The parameters of the ion source were as follows: spraying voltage 3.2 kV (for positive polarisation) and 2.2 kV (for negative polarisation), temperature 350 °C, shielding gas pressure 40 IU, auxiliary gas pressure 10 IU, capillary temperature 250 °C. Characteristic parameters for the identification of the tested analytes are given in Appendix A.

### 4.7. Statistical Analysis

Statistical analysis of the results was performed using Statistica 13 software (Statsoft, Carlsbad, CA, USA). One-way variance analysis (one-way ANOVA) was used to determine the significance of pathogen growth inhibition by *Trichoderma* and the degree of toxin biosynthesis inhibition. The significance of differences was determined at a significance level of α = 0.01. The homogeneity of the groups was determined using the Tukey HSD test.

### 4.8. Method Validation

In the analytical method of mycotoxin determination, the analytical range, the coefficient of determination (R2) for the calibration curve, the limit of detection (LOD) and the limit of quantification (LOQ) were determined for each analyte (Appendix A). The limits of detection and quantification were taken as analyte concentrations at which the signal-to-noise (S/N) ratio was equal to 3 and 10, respectively. Recovery (R%) and repeatability (expressed as relative standard deviation RSD%) were determined for each tested analyte. As the study analysed samples of post-culture media in which extreme levels of mycotoxins were found, it is virtually impossible to determine the recovery and precision parameters for such a wide range of analyte concentrations and levels. Therefore, amplifications in a limited range of mycotoxin concentrations were carried out on diluted extracts of the rice medium. The amplification levels and the results for recovery and repeatability in the method are shown in Appendix A. The matrix effect was verified by an analysis based on the comparison of the slope coefficients of the calibration curves prepared by analysing the standards in the solvent and in the matrix extracted from the monoculture medium containing the *T*. *atroviride* AN705 isolate. Recovery values for the analysed mycotoxins ranged from 70% to 112%, depending on the analyte and the level of amplification, with a relative standard deviation of 2.0% to 21.7%.

## 5. Conclusions

The results provided new information related to the ability of *Trichoderma* to modify *Fusarium* mycotoxins. The *Trichoderma* fungi not only showed the ability to inhibit pathogen growth and reduce toxin biosynthesis by *Fusarium* but also contributed to the glycosylation of DON and the biotransformation of ZEN to its hydroxylated derivative β-ZOL. We have also indicated that *Trichoderma* fungi may be a potential natural fungicide for *Fusarium* pathogens; however, their efficacy may vary between species and strains, and it is very likely to be different under in vivo conditions, as many environmental and agrotechnical factors may affect the antagonism efficacy. Therefore, further research in this area is required.

## Figures and Tables

**Figure 1 molecules-27-08146-f001:**
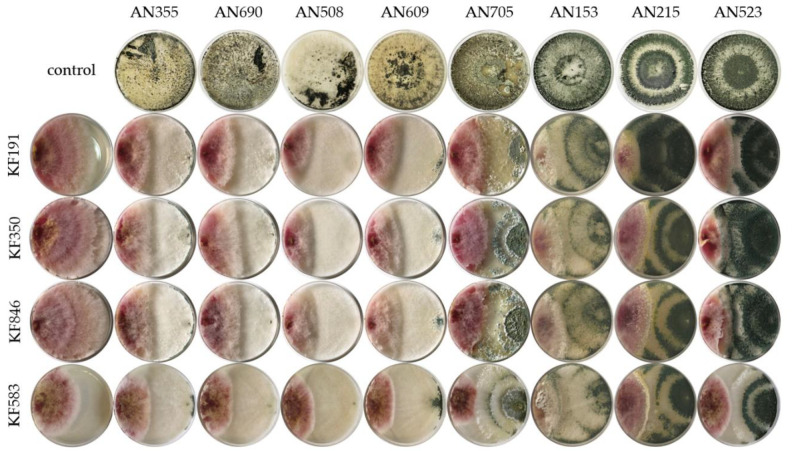
Fungal morphology of *Fusarium* in dual-culture assay on PDA after 5 days of incubation. *F*. *culmorum* KF191; *F*. *culmorum* KF350; *F*. *culmorum* KF846; *F*. *cerealis* KF583. *Fusarium* strain and *Trichoderma* strain grown alone-control, *Fusarium* strain with *T*. *viride* AN355, *T*. *viride* AN690, *T*. *viridescens* AN508, *T*. *viridescens* AN609, *T*. *atroviride* AN705, *T*. *atroviride* AN153, *T*. *atroviride* AN215, *T*. *atroviride* AN523.

**Figure 2 molecules-27-08146-f002:**
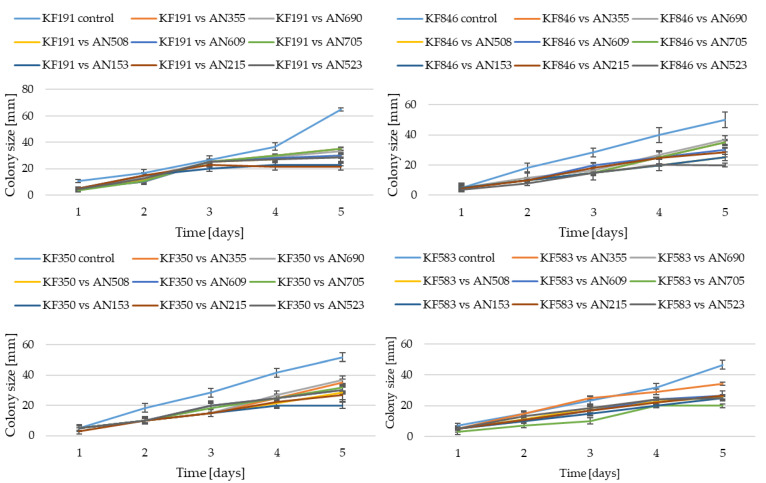
The inhibitory effect of antagonistic *Trichoderma* isolates on the growth rate of *Fusarium* spp. on PDA. The strain designations are the same as in Figure 1.

**Table 1 molecules-27-08146-t001:** Estimated inhibition (in % ± SD) of mycelial growth of *Fusarium* spp. by *Trichoderma* spp. after 5 days of co-incubation on PDA medium (*n* = 3).

Pathogen/ Antagonist		*F*. *culmorum* KF191	*F*. *culmorum* KF350	*F*. *culmorum* KF846	*F*. *cerealis* KF583	Average
*T*. *viride*	AN355	45.8 * ± 2.4	32.7 * ± 1.9	30.0 * ± 2.0	23.9 ± 2.1	33.1 ± 9.2
	AN690	53.6 * ± 2.4	42.3 * ± 9.6	48.0 * ± 2.0	45.1 * ± 2.1	47.3 ± 4.9
*T*. *viridescens*	AN508	49.0 * ± 3.9	31.4 * ± 7.8	28.0 * ± 7.2	50.0 * ± 15.0	39.6 ± 11.5
	AN609	51.6 * ± 1.6	42.3 * ± 3.8	36.0 * ± 2.0	42.3 * ± 5.3	43.0 ± 6.4
*T*. *atroviride*	AN705	56.8 * ± 3.9	42.9 * ± 2.9	59.3 * ± 5.0	49.3 * ± 7.6	52.1 ± 7.4
	AN153	45.3 * ± 1.6	39.1 * ± 5.6	50.0 * ± 2.0	57.7 * ± 4.2	48.0 ± 7.9
	AN215	63.0 * ± 3.3	59.6 * ± 1.9	50.0 * ± 2.0	42.3 * ± 5.3	53.7 ± 9.4
	AN523	65.6 * ± 4.1	49.4 * ± 6.2	42.7 * ± 6.4	42.3 * ± 5.3	50.0 ± 10.9

SD—standard deviation, the “*” symbol indicates a statistically significant result.

**Table 2 molecules-27-08146-t002:** The assessment of the interaction between *Trichoderma* vs. *Fusarium* colonies after 5 days of co-culturing on PDA medium.

Pathogen/ Antagonist		*F*. *culmorum* KF191	*F*. *culmorum* KF350	*F*. *culmorum* KF846	*F*. *cerealis* KF583	Average *
*T*. *viride*	AN355	8	4	4	4	5.0
	AN690	8	4	4	6	5.5
*T*. *viridescens*	AN508	8	4	6	6	6.0
	AN609	8	6	6	6	6.5
*T*. *atroviride*	AN705	8	6	4	6	6.0
	AN153	8	6	6	6	6.5
	AN215	8	6	6	6	6.5
	AN523	8	6	8	6	7.0

*—triplicates, all observations were equal in all cases and were averaged to identify the most effective antagonist.

**Table 3 molecules-27-08146-t003:** Mycotoxin production (mg/kg) by *Fusarium* strains on solid substrates (rice kernels) in the presence/absence of the *Trichoderma* strains.

Strain	Mycotoxin [mg/kg]								
	DON	DON3G	3- + 15-AcDON	NIV	FUS-X	ZEN	ZEN-14S	α-ZOL	β-ZOL
*F*. *culmorum* KF191									
KF191 control	<LOQ	<LOQ	23.95 ^a^ ± 3.89	<LOQ	<LOQ	47.36 ^a^ ± 8.69	3608 ^a^ ± 185	0.50 ^a^ ± 0.10	1.28 ^a^ ± 0.23
KF191 vs. AN355	<LOQ	<LOQ	2.85 ^b^ ± 0.93 (↓88%)	<LOQ	<LOQ	8.44 ^b^ ±1.64 (↓82%)	90.78 ^b^ ± 14.10 (↓97%)	0.18 ^b^ ± 0.04 (↓64%)	1.05 ^a^ ± 0.06 (↓18%)
KF191 vs. AN690	<LOQ	<LOQ	0.18 ^b^ ±0.02 (↓99%)	<LOQ	<LOQ	4.2 ^b^ ± 0.31 (↓91%)	19.62 ^b^ ± 4.36 (↓99%)	<LOQ ^b^ (↓100%)	0.41 ^b^ ± 0.11 (↓68%)
KF191 vs. AN508	<LOQ	<LOQ	0.43 ^b^ ± 0.09 (↓98%)	<LOQ	<LOQ	1.69 ^b^ ± 0.22 (↓96%)	10.53 ^b^ ± 2.85 (↓99%)	<LOQ ^b^ (↓100%)	0.20 ^b^ ± 0.08 (↓84%)
KF191 vs. AN609	<LOQ	<LOQ	0.4 ^b^ ±0.14 (↓98%)	<LOQ	<LOQ	0.56 ^b^ ± 0.05 (↓99%)	1.78 ^b^ ± 0.65 (↓99%)	<LOQ ^b^ (↓100%)	0.07 ^b^ ± 0.02 (↓94%)
KF191 vs. AN705	<LOQ	<LOQ	1.68 ^b^ ± 0.61 (↓93%)	<LOQ	<LOQ	1.97 ^b^ ± 0.21 (↓96%)	2.89 ^b^ ± 0.61 (↓99%)	<LOQ ^b^ (↓100%)	0.30 ^b^ ± 0.01 (↓76%)
KF191 vs. AN153	<LOQ	<LOQ	0.11 ^b^ ± 0.01 (↓99%)	<LOQ	<LOQ	0.99 ^b^ ± 0.17 (↓98%)	13.06 ^b^ ± 2.35 (↓99%)	<LOQ ^b^ (↓100%)	0.22 ^b^ ± 0.02 (↓83%)
KF191 vs. AN215	<LOQ	<LOQ	<LOQ ^b^ (↓100%)	<LOQ	<LOQ	2.75 ^b^ ± 0.49 (↓94%)	52.51 ^b^ ± 9.65 (↓98%)	<LOQ ^b^ (↓100%)	0.17 ^b^ ± 0.02 (↓84%)
KF191 vs. AN523	<LOQ	<LOQ	0.14 ^b^ ± 0.03 (↓99%)	<LOQ	<LOQ	28.21 ^b^ ± 9.73 (↓40%)	85.91 ^b^ ± 20.09 (↓97%)	<LOQ ^b^ (↓100%)	0.69 ^b^ ± 0.21 (↓46%)
*F*. *culmorum* KF350									
KF350 control	<LOQ	<LOQ	<LOQ	15.83 ^a^ ± 5.14	86.86 ^a^ ± 21.90	12.20 ^a^ ± 3.21	641.68 ^a^ ± 165	0.28 ^a^ ± 0.05	0.62 ^a^ ± 0.11
KF350 vs. AN355	<LOQ	<LOQ	<LOQ	2.01 ^b^ ± 0.54 (↓87%)	5.18 ^b^ ± 0.91 (↓94%)	7.59 ^b^ ± 1.70 (↓38%)	33.15 ^b^ ± 10.10 (↓95%)	0.22 ^a^ ± 0.05 (↓21%)	1.09 ^b^ ± 0.38 (↑75%)
KF350 vs. AN690	<LOQ	<LOQ	<LOQ	0.58 ^b^ ± 0.21 (↓96%)	0.55 ^b^ ± 0.11 (↓99%)	2.34 ^b^ ± 0.58 (↓81%)	35.23 ^b^ ± 15.53 (↓95%)	<LOQ (↓100%)	0.21 ^b^ ± 0.06 (↓66%)
KF350 vs. AN508	<LOQ	<LOQ	<LOQ	1.41 ^b^ ± 0.10 (↓91%)	0.41 ^b^ ± 0.05 (↓99%)	1.97 ^b^ ± 0.17 (↓84%)	52.78 ^b^ ± 6.27 (↓92%)	<LOQ (↓100%)	0.40 ^a^ ± 0.14 (↓35%)
KF350 vs. AN609	<LOQ	<LOQ	<LOQ	1.13 ^b^ ± 0.47 (↓93%)	0.78 ^b^ ± 0.25 (↓99%)	10.75 ^a^ ± 3.86 (↓12%)	17.31 ^b^ ± 4.89 (↓97%)	<LOQ (↓100%)	0.36 ^a^ ± 0.17 (↓42%)
KF350 vs. AN705	<LOQ	<LOQ	<LOQ	1.42 ^b^ ± 0.06 (↓91%)	3.19 ^b^ ± 1.24 (↓96%)	6.98 ^b^ ± 1.74 (↓43%)	31.13 ^b^ ± 6.39 (↓95%)	<LOQ (↓100%)	0.30 ^a^ ± 0.07 (↓52%)
KF350 vs. AN153	<LOQ	<LOQ	<LOQ	<LOQ ^b^ (↓100%)	0.48 ^b^ ± 0.19 (↓99%)	1.85 ^b^ ± 0.02 (↓85%)	123.38 ^b^ ± 11.18 (↓80%)	<LOQ (↓100%)	0.74 ^a^ ± 0.16 (↑19%)
KF350 vs. AN215	<LOQ	<LOQ	<LOQ	<LOQ ^b^ (↓100%)	0.13 ^b^ ± 0.04 (↓99%)	0.25 ^b^ ± 0.12 (↓98%)	22.01 ^b^ ± 8.31 (↓98%)	<LOQ (↓100%)	<LOQ ^b^ (↓100%)
KF350 vs. AN523	<LOQ	<LOQ	<LOQ	0.31 ^b^ ± 0.03 (↓98%)	0.54 ^b^ ± 0.07 (↓99%)	0.73 ^b^ ± 0.08 (↓94%)	57.61 ^b^ ± 13.94 (↓91%)	<LOQ (↓100%)	<LOQ ^b^ (↓100%)
*F*. *culmorum* 846									
KF846 control	82.53 ^a^ ± 14.96	<LOQ ^a^	111.09 ^a^ ± 33.53	<LOQ	<LOQ	164.29 ^a^ ± 16.77	2714 ^a^± 572	2.17 ^a^ ± 0.25	2.77 ^a^ ± 0.28
KF846 vs. AN355	12.29 ^b^ ± 1.00 (↓85%)	0.86 ^b^ ± 0.15	1.94 ^b^ ± 0.22 (↓98%)	<LOQ	<LOQ	64.33 ^b^ ± 5.96 (↓61%)	76.56 ^b^ ± 20.77 (↓97%)	1.09 ^b^ ± 0.42 (↓49%)	4.60 ^b^ ± 1.57 (↑66%)
KF846 vs. AN690	22.34 ^b^ ± 3.77 (↓73%)	2.43 ^b^ ± 0.21	3.26 ^b^ ± 0.90 (↓97%)	<LOQ	<LOQ	58.56 ^b^ ± 14.95 (↓64%)	49.20 ^b^ ± 6.84 (↓98%)	0.58 ^b^ ± 0.17 (↓73%)	2.42 ^a^ ± 0.91 (↓12%)
KF846 vs. AN508	18.40 ^b^ ± 1.13 (↓78%)	1.28 ^b^ ± 0.26	8.14 ^b^ ± 1.77 (↓92%)	<LOQ	<LOQ	40.56 ^b^ ± 11.57 (↓75%)	37.31 ^b^ ± 1.37 (↓98%)	0.68 ^b^ ± 0.25 (↓68%)	1.15 ^b^ ± 0.49 (↓58%)
KF846 vs. AN609	5.11 ^b^ ± 1.33 (↓94%)	1.55 ^b^ ± 0.48	0.81 ^b^ ± 0.19 (↓99%)	<LOQ	<LOQ	54.46 ^b^ ± 9.70 (↓67%)	46.04 ^b^ ± 6.17 (↓98%)	0.52 ^b^ ± 0.08 (↓76%)	1.56 ^a^ ± 0.14 (↓44%)
KF846 vs. AN705	6.21 ^b^ ± 0.49 (↓92%)	0.69 ^b^ ± 0.16	3.32 ^b^ ± 0.28 (↓97%)	<LOQ	<LOQ	104.30 ^b^ ± 16.69 (↓37%)	52.75 ^b^ ± 10.76 (↓98%)	0.33 ^b^ ± 0.05 (↓85%)	1.31 ^b^ ± 0.12 (↓53%)
KF846 vs. AN153	2.21 ^b^ ± 0.15 (↓97%)	1.41 ^b^ ± 0.34	1.64 ^b^ ± 0.23 (↓98%)	<LOQ	<LOQ	12.14 ^b^ ± 1.23 (↓93%)	13.89 ^b^ ± 2.19 (↓99%)	<LOQ ^b^ (↓100%)	0.42 ^b^ ± 0.09 (↓85%)
KF846 vs. AN215	1.90 ^b^ ± 0.60 (↓98%)	<LOQ ^a^	1.93 ^b^ ± 0.48 (↓98%)	<LOQ	<LOQ	6.89 ^b^ ± 2.28 (↓96%)	44.37 ^b^ ± 3.92 (↓98%)	<LOQ ^b^ (↓100%)	0.26 ^b^ ± 0.10 (↓91%)
KF846 vs. AN523	3.01 ^b^ ± 0.36 (↓96%)	<LOQ ^a^	0.38 ^b^ ± 0.03 (↓99%)	<LOQ	<LOQ	21.55 ^b^ ± 4.08 (↓87%)	24.83 ^b^ ± 5.64 (↓99%)	<LOQ ^b^ (↓100%)	0.29 ^b^ ± 0.05 (↓89%)
*F*. *cerealis* 583									
KF583 control	<LOQ	<LOQ	<LOQ	24.07 ^a^ ± 9.42	119.70 ^a^ ± 44.26	238.63 ^a^ ± 38.66	9337 ^a^ ± 1146	<LOQ	6.63 ^a^ ± 1.56
KF583 vs. AN355	<LOQ	<LOQ	<LOQ	0.84 ^b^ ± 0.29 (↓96%)	11.70 ^b^ ± 4.55 (↓90%)	2.20b ± 0.62 (↓99%)	4.20 ^b^ ± 0.72 (↓99%)	<LOQ	0.70 ^b^ ± 0.16 (↓89%)
KF583 vs. AN690	<LOQ	<LOQ	<LOQ	0.17 ^b^ ± 0.03 (↓99%)	11.98 ^b^ ± 0.82 (↓90%)	1.09 ^b^ ± 0.26 (↓99%)	3.41 ^b^ ± 0.34 (↓99%)	<LOQ	<LOQ ^b^ (↓100%)
KF583 vs. AN508	<LOQ	<LOQ	<LOQ	0.25 ^b^ ± 0.07 (↓98%)	1.54 ^b^ ± 0.07 (↓98%)	0.54 ^b^ ± 0.09 (↓99%)	1.31 ^b^ ± 0.47 (↓99%)	<LOQ	<LOQ ^b^ (↓100%)
KF583 vs. AN609	<LOQ	<LOQ	<LOQ	0.17 ^b^ ± 0.07 (↓99%)	0.87 ^b^ ± 0.03 (↓99%)	0.31 ^b^ ± 0.12 (↓99%)	0.67 ^b^ ± 0.11 (↓99%)	<LOQ	0.83 ^b^ ± 0.19 (↓87%)
KF583 vs. AN705	<LOQ	<LOQ	<LOQ	0.83 ^b^ ± 0.11 (↓96%)	9.84 ^b^ ± 0.76 (↓91%)	15.86 ^b^ ± 3.39 (↓93%)	9.70 ^b^ ± 1.34 (↓99%)	<LOQ	0.30 ^b^ ± 0.06 (↓95%)
KF583 vs. AN153	<LOQ	<LOQ	<LOQ	<LOQ ^b^ (↓100%)	<LOQ ^b^ (↓100%)	0.37 ^b^ ± 0.08 (↓99%)	1.50 ^b^ ± 0.38 (↓99%)	<LOQ	<LOQ ^b^ (↓100%)
KF583 vs. AN215	<LOQ	<LOQ	<LOQ	<LOQ ^b^ (↓100%)	<LOQ ^b^ (↓100%)	1.52 ^b^ ± 0.54 (↓99%)	7.14 ^b^ ± 0.57 (↓99%)	<LOQ	<LOQ ^b^ (↓100%)
KF583 vs. AN523	<LOQ	<LOQ	<LOQ	<LOQ ^b^ (↓100%)	0.31 ^b^ ± 0.11 (↓99%)	0.37 ^b^ ± 0.05 (↓99%)	1.62 ^b^ ± 0.34 (↓99%)	<LOQ	0.08 ^b^ ± 0.02 (↓98%)

↑ rise in mycotoxin concentration; ↓ decline in mycotoxin concentration; values within columns followed by the same letter are not significantly different according for α = 0.01.

## Data Availability

The data presented in this study are available upon reasonable request from the corresponding author.

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
