# Peer review of "Trichoderma versus Fusarium—Inhibition of Pathogen Growth and Mycotoxin Biosynthesis"

_molecules, 2022, doi:10.3390/molecules27238146_

Round 1
Reviewer 1 Report
Apart from testing novel Trichoderma and Fusarium strains, the results presented are largely a duplicate of a previous paper by the same first author (Suppressive effect of Trichoderma spp. on toxigenic Fusarium species. Polish Journal of Microbiology 2017, 66(1). DOI:10.5604/17331331.1234996), with the addition that the latter is far better organized. Because of this, the authors should cite this paper in the introduction and explain why they deemed it appropriate to expand the investigation to further strains. Likewise, the Discussion section should better emphasize the novel insight produced. In the present form, the manuscript is quite verbose and repetitive, and could be extensively shortened with no loss of information.
Further observations:
Because the abstract is of special importance, I suggested some changes to make it clearer and more effective.
Lines 89-100: summarize the observations in one or two short sentences, referring the reader to Figs 1 and 2 and table 1 for precise values. The names of species and genera must be in italics throughout.
Figure 1: indicate the type of medium and duration of incubation
Fig. 2: indicate the type of medium
Lines 262-263: combine into a single sentence
Lines 309-310: what do the authors mean by “pathogen increase”? Ref 31 does not seem to give any clue on this regard.
Further comments are in the attached pdf file.

Author Response
Responses
On our own behalf and of the other authors, we would like to thank the reviewers for their valuable comments and tips, which allowed us to further improve the manuscript. We are pleased that this manuscript has aroused your interest. Below we present responses to your remarks and comments.
Yours sincerely,
Marta Modrzewska
Reviewer 1
Q1 Apart from testing novel Trichoderma and Fusarium strains, the results presented are largely a duplicate of a previous paper by the same first author (Suppressive effect of Trichoderma spp. on toxigenic Fusarium species. Polish Journal of Microbiology 2017, 66(1). DOI:10.5604/17331331.1234996), with the addition that the latter is far better organized.
Re: Thank you for your opinion. Research methods are similar to those described in the cited article and this work was cited in the current manuscript. However, this manuscript emphasizes the potential of fungal enzymes to biosynthesis/transform mycotoxins into their modified forms. This is a very interesting observation that requires further research as knowledge about the potential of Trichoderma to transform mycotoxins is very limited. Because new strains of fungi were used in the research, the research carried out in the context of the inhibitory potential of Trichoderma spp. and mycotoxin transformation would be incomplete without assessing the antagonistic properties.
Q2 Because of this, the authors should cite this paper in the introduction and explain why they deemed it appropriate to expand the investigation to further strains.
Re: „Biocontrol measures are currently underexplored in the context of limiting Fusarium growth in cereal crops. Therefore, it is necessary to pre-select different strains of individual Trichoderma spp. species to assess their antagonistic potential." Introduction completed
Q3 In the present form, the manuscript is quite verbose and repetitive, and could be extensively shortened with no loss of information.
Re: Some parts of the text that in our opinion, were unnecessary have been removed.
Q4 Further observations:
Because the abstract is of special importance, I suggested some changes to make it clearer and more effective.
Re: Corrected.
Q5 Lines 89-100: summarize the observations in one or two short sentences, referring the reader to Figs 1 and 2 and table 1 for precise values. The names of species and genera must be in italics throughout.
Re: Italics have been used and redundant information has been removed, refer to tables and figures.
Q6 Figure 1: indicate the type of medium and duration of incubation
Re: Indicated in the text.
Q7 Fig. 2: indicate the type of medium
Re: Indicated in the text.
Q8 Lines 262-263: combine into a single sentence
Re: combined into one sentence
Q9 Lines 309-310: what do the authors mean by “pathogen increase”? Ref 31 does not seem to give any clue in this regard.
Re: “where Rc is the estimates of radial growth of a pathogen in control sample and R is the estimates of radial growth of a pathogen in the bi-culture” edited to be more clear
Q10 Further comments are in the attached pdf file.
Re: Thank you for all your advice and comments.
Reviewer 2 Report
Overall assessment
The research has been well planed and conducted. The results are also carefully described and statistically assessed. As the authors mention in the discussion similar work has been performed (various Trichoderma vs various Fusarium species), which give similar results. The research may therefore not provide extensive new knowledge, but I consider this to be a valuable addition for the community.
My major critique is that the authors did not expand the metabolite analysis to go beyond trichothecenes and zearalenones. Several other metabolites could have been interesting to analyze; for instance, aurofusarin would be very interesting as some of the Fusarium colonies seems to have a more intense red color when interacting with the Trichoderma strains.
Given that aurofusarin is a commercially available compound for which LC-MS/MS method have been described, I would like this to be further examined.
Minor comments
For Figure 1 it would interesting to see how the Trichoderma strains grow without the presence of Fusarium (an extra line of figures as control). Please add
The qualitative assessment needs to be better explained. It is difficult to understand how the data was generated
On Figure 2, the y-axis needs to be renamed. Size of Fusarium strain can be misleading. Colony size, radial growth or similar will be preferable
The manuscript would benefit greatly from a thorough check to eradicate spelling and grammatical errors. Check that all names of organisms are in italic and rephrase some sentences (Line 224)
Author Response
Responses
On our own behalf and of the other authors, we would like to thank the reviewers for their valuable comments and tips, which allowed us to further improve the manuscript. We are pleased that this manuscript has aroused your interest. Below we present responses to your remarks and comments.
Yours sincerely,
Marta Modrzewska
Reviewer 2
Q1 My major critique is that the authors did not expand the metabolite analysis to go beyond trichothecenes and zearalenones. Several other metabolites could have been interesting to analyze; for instance, aurofusarin would be very interesting as some of the Fusarium colonies seems to have a more intense red color when interacting with the Trichoderma strains.
Given that aurofusarin is a commercially available compound for which LC-MS/MS method have been described, I would like this to be further examined.
Re: Thank you for your valuable comments. In our research, we focused on toxins from the trichothecenes group, zearalenone, and their modified forms. Our attention was due to the fact that these compounds are often observed in cereal grains and are regulated by European law (some trichothecenes and zearalenone). Indeed, the inclusion of aurofuzarin in research, perhaps, would allow us to obtain interesting research results. We will definitely take this into account in the future, as we continue to conduct research related to the effect of Fusarium fungi on the metabolome of antagonists (Trichoderma).
Q2 Minor comments
For Figure 1 it would interesting to see how the Trichoderma strains grow without the presence of Fusarium (an extra line of figures as control). Please add
Re: Added and a new figure was created.
Q3 The qualitative assessment needs to be better explained. It is difficult to understand how the data was generated
Re: This is a qualitative/visual assessment, but corrections have been made to the text
Q4 On Figure 2, the y-axis needs to be renamed. Size of Fusarium strain can be misleading. Colony size, radial growth or similar will be preferable
Re: axis name changed to “colony size”
Q5 The manuscript would benefit greatly from a thorough check to eradicate spelling and grammatical errors. Check that all names of organisms are in italic and rephrase some sentences (Line 224)
Re: Italics have been used, and corrections have been made.
Round 2
Reviewer 2 Report
Thank you very much for the revised manuscript. I accept that you will not analyze for other metabolites in the current study (such as aurofusarin) and hopefully will include it in future experiments. The option only to focus on mycotoxins with established thresholds is understandable, but it also means that you might be missing out on important information (perhaps some metabolites are enhanced many times during the interactions?). Therefore, I highly recommend that you include these possibilities in the discussion.
Author Response
Re: suggestions followed, changes made, line 217-223